# Activation of TGF-β Canonical and Noncanonical Signaling in Bovine Lactoferrin-Induced Osteogenic Activity of C3H10T1/2 Mesenchymal Stem Cells

**DOI:** 10.3390/ijms20122880

**Published:** 2019-06-13

**Authors:** Yixuan Li, Wei Zhang, Fazheng Ren, Huiyuan Guo

**Affiliations:** 1Beijing Advanced Innovation Center for Food Nutrition and Human Health, College of Food Science and Nutritional Engineering, China Agricultural University, Beijing 100083, China; liyixuan9735@126.com (Y.L.); zw_7431@163.com (W.Z.); renfazheng@cau.edu.cn (F.R.); 2Beijing Laboratory of Food Quality and Safety, China Agricultural University, Beijing 100083, China; 3Key Laboratory of Functional Dairy, co-constructed by Ministry of Education and Beijing Government, China Agricultural University, Beijing 100083, China

**Keywords:** Lactoferrin, osteogenic, mesenchymal stem cells, TGF-β/Smad2 signaling pathway

## Abstract

Lactoferrin (LF) is known to modulate the bone anabolic effect. Previously, we and others reported that the effects of LF on the bone may be conferred by the stimulation of transforming growth factor β (TGF-β) signaling in the preosteoblast. However, the underlying molecular mechanisms of LF-induced osteogenic differentiation of mesenchymal stem cells (MSCs) has not been identified. In this study, we tested the hypothesis that the effects of LF on osteogenesis of MSCs required mediation by TGF-β Receptors and activating TGF-β signaling pathway. Using siRNA silencing technology, the knockdown of TGF-β Receptor II (TβRII) could significantly attenuate LF’s effect on the proliferation rate and alkaline phosphatase (ALP) activity of MSCs. It indicated that LF induced osteogenic activity that is dependent on TβRII in C3H10T1/2. Subsequently, it was shown that LF activated Smad2. Downregulating TGF-β Receptor I (TβRI) with SB431542 attenuated the expression of p-Smad2 and p-P38, also the LF-induced the osteogenic activity. Besides, the stimulation by LF on the expression of Osteocalcin (OCN), Osteopontin (OPN), Collagen-2a1 (Col2a1), and Fibroblast Growth Factor 2 (FGF2) were abolished by SB431542. These results confirmed that LF induced osteogenic activity though the TGF-β canonical and noncanonical signaling pathway. This study provided the first evidence of the signaling mechanisms of LF’s effect on osteogenesis in MSCs.

## 1. Introduction

Lactoferrin (LF) is an 80-kDa iron-binding glycoprotein that is present in colostrum, milk, serum, and mucosal fluids [1,2]. The original study of LF found that it has antimicrobial activity and functions as a regulator of immune response [3,4]. In recent years, several studies have shown that LF acting as a growth factor is crucial in the development of skeletal system in vivo and in vitro [5,6,7,8,9]. Oral LF administered to Ovx rats effectively prevented Ovx-induced bone loss and improved bone mineral density [10]. In vitro, LF has also been demonstrated to promote osteoblast-like cells proliferation and differentiation [7,8]. In addition, in a clinical study, LF significantly reduced bone resorption markers, when postmenopausal women randomly received oral RNAse-enriched LF [11]. LF is widely used as a functional ingredient in infant formulas because of these functions [12,13,14,15].

The transforming growth factor-beta (TGF-β) have widely recognized roles in bone formation during mammalian development in the body [16,17]. Autocrine and paracrine stimulation by TGF-β is important in the maintenance and expansion of mesenchymal stem cells [18]. TGF-β signaling also promotes osteoprogenitor proliferation and early differentiation [19,20,21]. Furthermore, it was reported that TGF-β signaling commitment to the osteoblastic lineage through selective TGF-β canonical and noncanonical pathways [22,23]. LF was shown to promote osteogenesis through the activation of canonical TGF-β signaling in preosteoblastic cell line MC3T3-E1 and C57BL/6J mice [24]. Besides, TGF-β noncanonical pathways play an important role during bone formation that is activated by LF. Grey et al. [25] reported that LF induces osteoblast growth by activating the Mitogen-activated protein kinase (MAPK)–extracellular signal regulated kinase (ERK) 1/2. Zhang et al. found that LF mainly stimulated osteoblast differentiation through the P38 MAPK signaling pathway [26]. Notwithstanding the above studies, TGF-β canonical and noncanonical pathways have been shown to be involved in the LF-induced osteoblastic lineage. However, the functional TGF-β signaling specific for LF-induced osteogenic activity of mesenchymal stem cells (MSCs) still remains to be elucidated.

The mesenchymal stem cells (MSCs), which are the progenitors of osteoblasts, are a multipotent cell type that can differentiate into the osteoblastic, chondrogenic, myogenic, or adipogenic lineages, which was considered as the progenitor cells in bone formation. C3H10T1/2 cells, which are a murine multipotent mesenchymal precursor cell line, have been used as a stem cell model as well as a model for osteogenic and chondrocytic differentiation [27]. In the present study, the TGF-β receptor has been confirmed to involved in BMP-9-induced osteogenic differentiation of C3H10T1/2 MSCs [28]. Furthermore, retinoic acid has inhibitory effects on BMP4 induction of C3H10T1/2 adipocytic commitment via downregulating Smad/p38 MAPK signaling [29]. These results suggested that the TGF-β canonical and noncanonical pathway may play a functional role in the differentiation of C3H10T1/2 cells. However, whether TGF-β signaling pathway is involved in LF-induced osteogenic differentiation in C3H10T1/2 MSCs is not clear.

The purpose of this study was to explore whether LF could stimulated TGF-β/Smad2 signaling pathway and induce the osteogenic differentiation of MSCs. To test our hypothesis, C3H10T1/2 was treated with 100 μg/mL LF, and the proliferation and differentiation of MSCs were detected. Furthermore, to prove that LF stimulated TGF-β receptors II–mediated Smad2 signaling and upregulates the expression of growth factors to induce MSCs differentiate into the osteoblast, the expression of TGF-β receptors and the activation of Smad2 were measured. In addition, we sought to determine the crosstalk of P38 pathway and Smad2 signaling, to show that LF activates the TGF-β noncanonical signaling pathway in the osteogenic differentiation of MSCs.

## 2. Results

### 2.1. TGF-β Pathway Is Required for LF-Induced Proliferation and Differentiation in Mesenchymal Stem Cells C3H10T1/2

LF has been reported to stimulate proliferation and differentiation in osteoblasts [5]. As a precursor to osteoblast, we stimulated C3H10T1/2 cells with LF at concentrations from 1 to 1000 μg/mL and measured cell proliferation rates and the ALP activity, as assessed, to further address whether LF has an effect on the proliferation and differentiation of mesenchymal stem cells. We found that the C3H10T1/2 cells exhibited a significant increase in the proliferation rate when an LF dose >10 μg/mL (Figure 1A). Additionally, the treatment of LF also increased the ALP activity when the dose >1 μg/mL (Figure 1B). Thus, in general, LF promotes the proliferation and differentiation of mesenchymal stem cells. It has been previously shown that TGF-β isoforms stimulated cell proliferation and bone differentiation in C3H10T1/2 cells [30]. That means the effects of LF were similar to that of the TGF-β isoforms treatment. Therefore, we speculated whether the TGF-β/Smads pathway plays the important role in LF-induced C3H10T1/2 cells proliferation and differentiation. We used SB431542 to examine whether the TGF-β pathway involved in LF-induced proliferation and differentiation in C3H10T1/2 cells, because SB431542 can inhibit the TGF-β pathway by abolishing the phosphorylation of TGF-β receptor I. As shown in Figure 1C, LF treatment increased proliferation rate of 100mg/mL in C3H10T1/2 cells, it was reduced by up to 67% in the presence of SB431542. Likewise, LF (100 μg/mL) significantly increased the ALP activity in cells, and SB431542 reduced LF-induced ALP activity by up to 80%. (Figure 1D). These results showed that the TGF-β pathway is required for LF-induced osteoblast proliferation and differentiation in mesenchymal stem cells.

### 2.2. TGF-β Pathway Is Involved in the LF-Induced Expression of Gene Markers for the Osteogenic Activity in Mesenchymal Stem Cells

C3H10T1/2 cells are multipotent mesenchymal fibroblasts that are able to differentiate along osteogenic, chondrogenic, adipogenic, or myogenic lineages, depending on culture conditions. We demonstrated that the C3H10T1/2 cells were associated with the osteogenic activity of LF. To further prove the role of LF in enhancing expression of gene markers for the osteogenic activity in C3H10T1/2 cells, we used quantitative real-time PCR to quantitate the expression levels of osteocalcin (*Ocn*), osteopontin (*Opn*), collagen-2a1 (*Col2a1*), and fibroblast growth factor 2 (*Fgf2*). These osteoblastic markers are actively expressed and produced to facilitate bone formation [27,28]. As expected, all of the osteoblastic gene markers exhibited significant upregulation in their expression after LF treatment (Figure 2). When the cells were grown in medium supplemented with SB431542 before LF treatment, the increase in expression of these gene markers was significantly hindered. The results further indicated that the TGF-β signaling pathway plays an important role in the LF-induced osteogenic activity in mesenchymal stem cells.

### 2.3. LF Induces Osteogenic Activity in Mesenchymal Stem Cells Is Mediated by TGF-β Receptors

The TβRII is constitutively active kinase in Canonical TGF-β signaling, which phosphorylates TβRI upon ligand binding. The mRNA and protein levels of TGF-β receptor I and II were determined in LF-treated mesenchymal stem cells to determine whether TGF-β receptors are involved in the osteogenic activity following LF stimulation. LF treatment significantly increased the levels of TβRI and TβRII. Time-course analysis of LF-induced activation showed that the LF-induced mRNA expression of TβRI and TβRII peaked at 12 h posttreatment, and then declined at 24 h (Figure 3A,B). The protein levels of TβRI were increased at 4 h by LF and continued to increase at 12 h post-stimulation, and then gradually declined at 24 h. In addition, the expression of TβRII was induced at 12 h by LF and continued to increase at 24 h post-stimulation, peaking at 24 h posttreatment (Figure 3C,D). It indicated that the expression of protein was followed by mRNA expression.

It is of great interests to explore the role of TβRII in LF-induced anabolic effect in C3H10T1/2 cells, we proceeded to abolish TβRII in these cells to investigate the effect of LF treatment because the expression of TGF-β receptors have been shown increased by LF. We tested TβRII expression was effectively silenced by TβRII siRNA. The deficiency of TβRII expression was confirmed by Western blotting, which showed that TβRII expression was suppressed well by siRNA, which, as expected, led to a reduction in the expression of p-Smad2 (Figure 4A). Next, we evaluated the osteoinductive effect of LF in TβRII knockdown cells. The results of MTT and ALP assays showed that the silencing of TβRII significantly attenuated LF’s effect on proliferation rate and ALP activity by up to 50% in MC3T3-E1 cells (Figure 4B,C), which suggested that T*β*RII could be involved in LF-induced osteogenic activities in mesenchymal stem cells.

### 2.4. LF Activates TGF-β/smad2 Signaling Pathway in Mesenchymal Stem Cells

TGF-β signaling has been reported to be an important pathway in LF—regulating the growth of bone cells. In 2009, Brandl found that LF induced TGF-β receptor activation and nuclear Smad2 translocation to stimulate the proliferation of chondrocytes [21]. Additionally, Li et al. demonstrated that LF promotes osteogenesis through TGF-β Receptor II binding activation of canonical TGF-β signaling in MC3T3-E1 cells and C57BL/6J Mice [24]. Thus, it was plausible to propose that LF-induced proliferation and differentiation in C3H10T1/2 cells could be through the activation of TGF-β/Smad2 signaling. Our results showed that the levels of phosphorylation and nuclear translocation of Smad2 are induced at 15 min. by LF and thy continue to increase at 180 min. post-stimulation, peaking at ∼60 min. posttreatment, and then gradually declining. In addition, there was no significant difference in the levels of Smad4 induction between all the treating groups (Figure 5A,B). C3H10T1/2 cells were stimulated with LF in the presence of inhibitor SB431542 to investigate the basis of the increased steady-state levels of p-Smad2 expression and nuclear translocation. The levels of p-Smad2 and nuclear translocation are decreased by up to 100% with the addition of SB431542 (Figure 5C,D), implying that LF-induced anabolic effect is TGF-β/Smad2 signaling pathway dependent in mesenchymal stem cells.

### 2.5. LF Activates TGF-β/smad2 Pathway Dependent on p38 MAPK Signaling Pathway

The previous studies have demonstrated that MAPK is highly capable of inducing osteogenic differentiation and bone formation [31]. In addition, it is known that the MAPK pathway is a TGF-β noncanonical pathway. Thus, we sought to determine whether there was an interaction between the TGF-β/smad2 pathway and MAPK pathway in LF-induced bone formation. First, the C3H10T1/2 cells were exposed to LF in the presence of SB203580, U0126, and SP600125, which are specific inhibitors of p38, ERK1/2, and JNK respectively. Afterwards, the expression of p-Smad2 was determined using Western blot. By treating C3H10T1/2 cells with inhibitors (SB’, U, SP), we found that the p38, ERK1/2, and JNK inhibitors did not suppress any LF-induced enhancement of Smad2 activation (Figure 6A). However, the inhibition of TβRI with SB431542 dramatically attenuated LF-induced P38 phosphorylation in C3H10T1/2 cells, implying that LF-induced P38 activation is TβRI dependent. Unlike P38, there was no significance effect to LF-induced ERK1/2 and JNK activation by TGF-β receptor I inhibitor SB431542, (Figure 6B). The results showed that LF activated P38 MAPK signaling dependent on the TGF-β/Smad2 signaling pathways in mesenchymal stem cells.

## 3. Discussion

LF, as a potent anabolic factor, has been thought to promote bone formation [5]. Bone formation involves the commitment of MSCs to the osteoblastic lineage for osteogenic differentiation [32]. The osteogenic activity of LF in MSCs remains largely unknown, although previous studies have explored the osteotropic effect of LF in preosteoblast MC3T3-E1. In addition, the molecular mechanisms underlying LF-induced osteotropic activity have been demonstrated to be related to the TGF-β signaling pathway [24]. However, the function of TGF-β receptor and the target protein of Smads is unclear in MSCs. In the study, we provided direct evidence that the MSC osteogenic activity, similar to preosteoblast, is able to stimulated by LF. Moreover, TGF-β/Smad2 signaling has been proved to be involved in the proliferation and differentiation of MSCs. To the best of our knowledge, this study is the first to provide evidence that LF activates TGF-β signaling for the osteotropic effect in MSCs; this work is a continuation and addition to our previous research.

In the process of bone regeneration, MSCs was the progenitor cell of osteogenic differentiation, which first differentiates into preosteoblasts and then into mature osteoblasts. New bone is formed by the synthesis of the extracellular bone matrix [32,33]. The proliferation and differentiation of osteoblast are governed by transcription factors, which result in the expression of phenotypic genes that lead to the acquisition of the osteoblast phenotype [34,35]. Foremost, the osteogenic differentiation of MSCs is characterized by the expression of the ALP activity. Subsequently, many growth factors are actively expressed and produced to facilitate bone formation, including osteocalcin (Ocn), osteopontin (Opn), collagen-2a1 (Col2a1), and fibroblast growth factor 2 (Fgf2) [36,37]. Pei-Lin Shao et al. showed that ALP, OCN, and collagen expression were promoted in osteogenic differentiation [38]. In addition, the levels of some osteoblastic markers (ALP, OCN, Col) were expressed relatively higher in cell proliferation and migration induced by retinoic acid in C3H10T1/2 cells [30]. Similarly, LF treatment also led to a similar pattern of increase in the expression of osteoblastic markers. In our previous work, ALP, OCN, OPN, collagen, and Fgf2 were demonstrated to be significantly promoted by LF during osteogenesis in preosteoblast [24]. However, we found that LF treatment increased the mRNA expression levels of these growth factors in MSCs in this study. Our study confirmed that osteogenic differentiation from progenitor cells to preosteoblasts by LF treatment could be induced in vitro; it provided complete evidence for the osteogenic activity of LF.

We have shown that LF significantly induces osteogenic differentiation of MSCs. However, little is known regarding the receptors underlying LF-induced osteogenesis of MSCs. In our current study, we show that TGF-β canonical signaling pathway plays an essential role in the LF-induced osteotropic factors, as expressed in the MC3T3-E1 cells. Thus, it might be possible that LF activates the osteogenic activity via TGF-β receptors in MSCs. To date, four different type II TGF-β receptors have been identified, including TGFβRII, BMPRII, ActRII, and ActRIIB. Several studies have demonstrated that type II TGF-β receptors are required for the osteogenic activity of BMPs [28,39]. In our previous study, LF fulfilled their signaling activity by interacting with receptor complexes that are composed of type I and type II TGF-β receptors in MC3T3-E1 [24]. Similarly, in C3H10T1/2 cells, we found that LF significantly induced the expression of TGF-β receptor II and receptor I, whereas the proliferation and differentiation of MSCs were inhibited upon knockdown of the expression of TβRII by siRNA and inhibition phosphorylation of TβRI by SB431542, suggesting that LF-induced proliferation and osteogenic differentiation in these cells are primarily mediated by the TGF-β receptors.

The TGF-β receptors often involve canonical and noncanonical signaling pathways, such as the signal effectors upon the activation of the osteotropic activity. In the study, we show that LF can function by interacting with TGF-β receptors to stimulate TGF-β canonical signaling. Thus, it might be possible that LF also activates the MAPK pathway via TGF-β receptors by way of a TGF-β noncanonical pathway. The physiological functions of MAPKs (mainly p38, ERK1/2, and JNKs) in osteogenic differentiation and bone formation have been deeply investigated both in vitro and in vivo [31,40,41,42,43,44,45,46,47]. However, the obtained results from the in vitro studies are controversial, with some studies suggesting a stimulatory role of MAPKs in osteogenic differentiation and bone formation [44,45,46,47], and others proposing that MAPKs is inhibitory [31,40,41,42]. The conflicting results of in vitro studies emphasize the need to explore the role of the MAPKs in osteogenic differentiation and bone formation by in vivo assay. In our study, we find that p38 activation was capable of stimulating osteoblast differentiation, however the ERK1/2 and JNK pathway did not show any activation in LF-induced osteotropic activity. Additionally, the activation of p38 was found to participate in BMP2-induced commitment of C3H10T1/2 MPCs to the adipocyte lineage [48]. These results affirmatively supported that p38 MAPKs also play positive roles in regulating osteogenic differentiation and bone formation. This is consistent with our previous findings that LF stimulated the osteotropic activity of MC3T3-E1 cells by inducing p38 activation [24,26]. In addition, growth factors have shown to trigger MAPKs in different cell models. MAPKs can also be activated by BMPs stimulation [40,41,48,49,50,51], which represents an important mechanism for the non-Smads pathway(s) of BMPs signaling. Similarly, we demonstrated that LF activates the p38 pathway via the TGF-β receptor by way of a TGF-β noncanonical pathway. However, TβRII knockdown by siRNA in MC3T3-E1 cells did not show any reduction in LF-induced p38 activation, nor was there any attenuation observed on p38 activation by the inhibition of TβRI activity [24]. The discrepancy between C3H10T1/2 cells and that of MC3T3-E1 cells might be due to the fact that, in the MAPK pathway, LF binds to different receptors in different cell types.

## 4. Materials and Methods

### 4.1. Materials

Bovine LF with 95% purity (SDS-PAGE) [52] was provided by the Australian Yosica Holding (Melbourne, Australia). Modified essential medium (MEM) and fetal bovine serum were obtained from Life Technologies (Rockville, MD, USA). Trypsin from porcine pancreas and DMSO were purchased from Sigma-Aldrich (St. Louis, MO, USA). Rabbit antibody against Smad2, Smad4, phospho-Smad2 (p-Smad2), anti-mouse, and rabbit immunoglobulin G (IgG) peroxidase conjugate antibodies (Cell Signaling Technology (Beverly, MA, USA). SB431542 was purchased from Calbiochem (Cambridge, MA, USA). Rat anti-TGFβRII antibody for western blot was purchased from R&D Systems (Minneapolis, MN, USA). Mouse anti-TGFβRII antibody for immunoprecipitation and goat anti-rat IgG-HRP antibody were obtained from Santa Cruz Biotechnology (Santa Cruz, CA, USA). Rabbit anti-histone H3 and β-actin antibodies were obtained from Beyotime (Nantong, China).

### 4.2. Cells Culture and Treatment

The mouse mesenchymal stem cells C3H10T1/2 were obtained from ATCC (Rockville, MD, USA) and subcultured at 60%–70% confluence in MEM containing 10% fetal bovine serum at 37 °C under 95% air/5% CO_2_ atmosphere. The C3H10T1/2 cells were plated in 60-mm dishes at density of 4 × 10^4^ cells/cm^2^ and grown to 90% confluence. The cells were stimulated with LF (100 μg/mL) in serum-free media to examine the downstream signaling pathway. The inhibitor were added 1 h before LF to assess the effects of kinase inhibitor on LF-induced cell activity. The LF dosage and treatment time are chosen according to our previous experiment [24].

### 4.3. Cell Proliferation Assay

The colorimetric 3-(4,5-dimethylthiazol-2-yl)-2,5-diphenyltetrazolium bromide (MTT; Sigma, St. Louis, MO, USA) assay was used to assess LF’s effect on proliferation of osteoblasts. The cells were seeded into 96-well tissue culture plates at a density of 2.5 × 10^4^ cells/cm^2^. Twenty-four hours after subculturing, the cells were changed to serum-free medium for a further 24 h before the addition of LF. Treatments were for a further 48 h. MTT (5 mg/mL in PBS) was added to each well for 4 h. Afterwards, the supernatant was removed and dimethylsulfoxide was added to solubilize MTT for 4 h. After extraction with dimethylsulfoxide, the optical density was measured at 495 nm (plate reader, Bio-Rad Laboratories, Hercules, CA, USA).

### 4.4. ALP Activity Assay

The cells were plated in 24-well plates at a density of 2 × 10^4^ cells/cm^2^ and the medium was changed every two days until the seventh day. For the inhibitor assay, at the last day, after SB431542 was added 1 h, the cells were stimulated with LF (100 μg/mL) for 24 h. Cells were then washed with physiological saline for three times and the cell layers were scraped and sonicated in 0.1 M Tris buffer (pH 7.4) containing 1% Triton X-100. Activity was quantitated in cell lysate using an ALP activity diagnostic kit (Roche Diagnostics GmbH, Basle, Swiss) with automated clinical chemistry analyzer (Hitachi, Tokyo, Japan). The aliquots of cell lysate were subjected to protein assay while using a bicinchoninic acid (BCA) protein assay kit (Pierce, Rockford, IL, USA) and the ALP activities were normalized to the total protein.

### 4.5. Western Blot

The cells were plated in six-well plates at a density of 4 × 10^4^ cells/cm^2^ and lysed in 100 mL of lysis buffer (10 mg/mL aprotinin, 10 mg/mL leupeptin, 10 mg/mL pepstatin, 10 mM iodoacetamide, and 1 mM PMSF) each well for 30 min. on ice. Nuclear extracts were prepared with a nuclear and cytoplasmic protein extraction kit. The protein concentration was determined using a BCA protein assay kit. For Western blotting, equal amounts of cleared cell lysates were separated by SDS-PAGE, followed by the transfer of proteins to the polyvinylidene difluoride (PVDF) membrane (Millipore, Temecula, CA, USA). After being blocked with 5% defatted milk and washed, the membrane was incubated with primary antibody, followed by incubation with horseradish peroxidase-conjugated secondary antibody. The washed blot was developed while using enhanced chemiluminescence reagent (Millipore) and was exposed to MulitImager (GE, AI600, Boston, MA, USA).

### 4.6. Quantitative Real-Time PCR Analysis

Total RNA was isolated from C3H10T1/2 using TRIzol (Invitrogen, Carlsbad, CA, USA) and reverse translated to cDNA using first-strand cDNA synthesis kit (Takara Biotechnology, Dalian, China). Quantitative RT-PCR was performed with SYBER Premix Ex Taq according to the company’s instructions (Takara Biotechnology, Dalian, China). Table 1 indicates the primer sets that were used for amplification. For each gene, the level of expression at the zero time point was normalized to 1 and all the other values were calculated relative to that point. The experiments were performed in duplicate, with three biological repeats for each experiment producing similar results.

The siRNA plasmids for TβRII were constructed by Sangon Biotech (Shanghai, China). The siRNAs were designed to target TβRII were GACCUCAAGAGCCUCUAACATT (sense), UGUUAGAGCUCUUGAGGUCTT (antisense). The C3H10T1/2 cells were transfected with appropriate plasmids expressing TβRII siRNA and control scramble siRNA while using Lipofectamine 2000 (Invitrogen, Carlsbad, CA, USA) and subjected to the Western blotting analysis, proliferation assay, and ALP assay.

### 4.7. Statistical Analysis

The results were expressed as means ± standard deviations. Data analysis was carried out while using SPSS software (version 17.0 SPSS Inc., Chicago, IL, USA). The evaluation of significance between groups was performed using One-factor analysis of variance (ANOVA), followed by Duncan’s post hoc test. Univariate two-factor ANOVA was used to analysis the main effects and interactions of two factors (LF × SB431542 (SB), LF × siRNA (si), LF × SB203580 (SB’), LF × U0126 (U), and LF × SP600125 (SP)). When the main effect was significant, mean comparisons were conducted by using a Tukey’s post hoc test. If an interaction between two factors was found, a mean comparison was made conditionally. Statistical significance was defined at a level of *p* < 0.05 in all tests.

## 5. Conclusions

In conclusion, we find that LF is shown to activate the TGF-β/Smad2 pathway in the induction of the osteogenic proliferation and differentiation of MSCs, which implies that TGF-β signaling is an essential regulatory component in LF osteoinductive activity. Notably, using specific inhibitor and siRNA for TGF-β receptor I and TGF-β receptor II, respectively, we find that LF-induced osteogenic activity of MSCs are significantly hindered in vitro. It provides new evidence for the receptor in osteoblast differentiation. In addition, LF activated the osteogenic proliferation and differentiation through the TGF-β/Smads canonical and noncanonical pathway, but the crosstalk of these pathways needs further study. This work perfects the molecular mechanisms underlying the anabolic effect of LF on bone and will promote the further use of LF as a natural infant formula reagent.

## Figures and Tables

**Figure 1 ijms-20-02880-f001:**
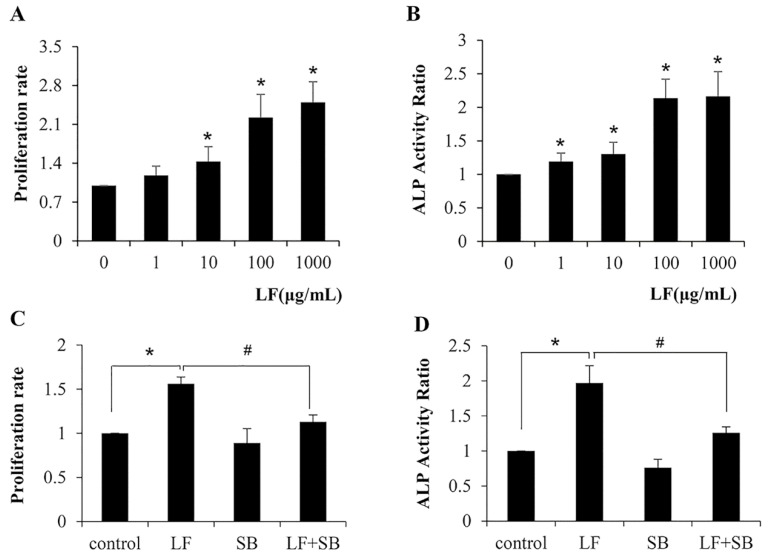
Lactoferrin (LF)-induced C3H10T1/2 cells proliferation and differentiation is inhibited by TGF-β inhibitor SB431542. C3H10T1/2 cells were treated with LF and/or SB, as described in Materials and Methods before they were subjected to proliferation and differentiation analysis. (**A**,**B**) The effect of LF treatment for indicated concentrations on mesenchymal stem cells (MSCs) proliferation and differentiation. (**C**,**D**) 100 μg/mL LF treatment in the presence of SB on MSCs proliferation and differentiation. Values are mean ± SDs, *n* = 3 (means of five replicates). (* *p* < 0.05 versus control group; # *p* < 0.05 LF-treated group versus LF+SB treated group).

**Figure 2 ijms-20-02880-f002:**
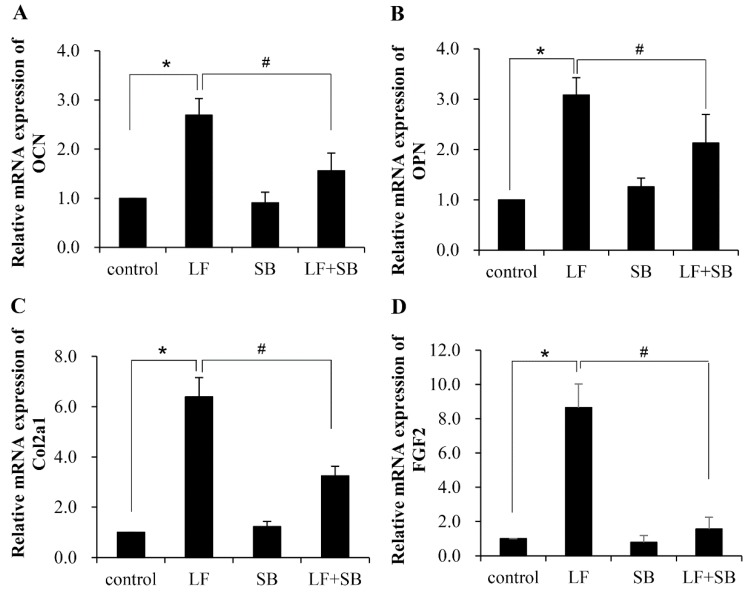
Transforming growth factor β (TGF-β) pathway is involved in LF-induced upregulation of osteogenic factors in C3H10T1/2 MSCs. C3H10T1/2 cells were pretreated with SB for one hour and followed by treatment with LF for four hours. Expression of OCN (**A**), OPN (**B**), Col2a1 (**C**), and FGF2 (**D**) were measured by Quantitative Real-Time PCR (q-RT-PCR). Values are mean ± SDs, *n* = 3 (means of five replicates). (* *p* < 0.05 versus control group; # *p* < 0.05 versus LF+SB treated group).

**Figure 3 ijms-20-02880-f003:**
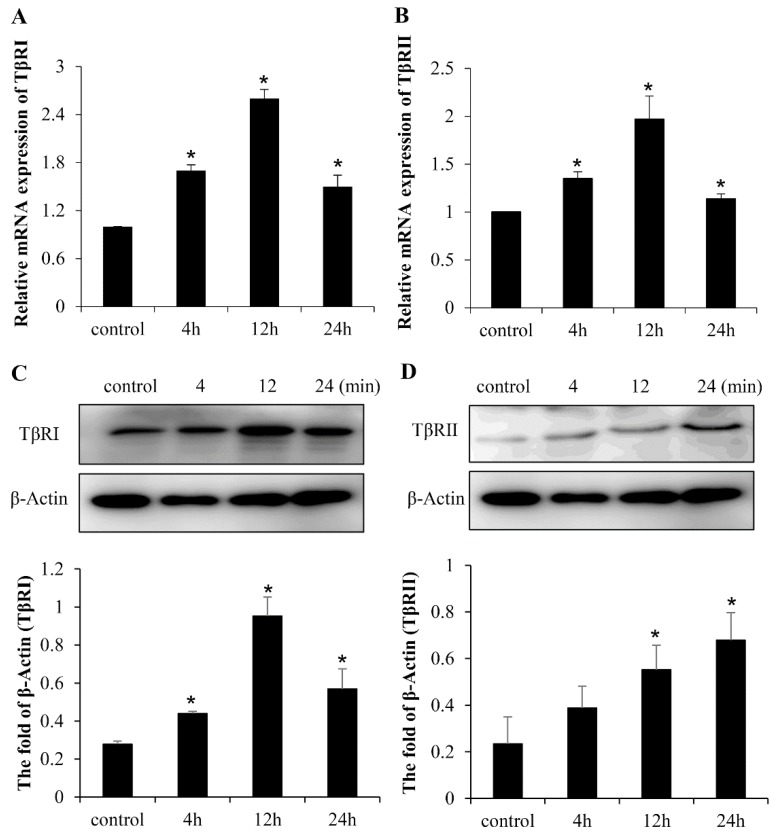
Effects of LF on the expression of TGF-β receptors in C3H10T1/2 cells. Time-course analysis of mRNA expression of TGF-β receptor I (**A**) and receptor II (**B**) after treatment with 100 μg /mL LF in MSCs. Time-course analysis of the expression of TGF-β receptor I (**C**) and receptor II (**D**) after treatment with 100 μg /mL LF on the protein level in MSCs. Values are mean ± SDs, *n* = 3 (means of five replicates). (* *p* < 0.05 versus control group).

**Figure 4 ijms-20-02880-f004:**
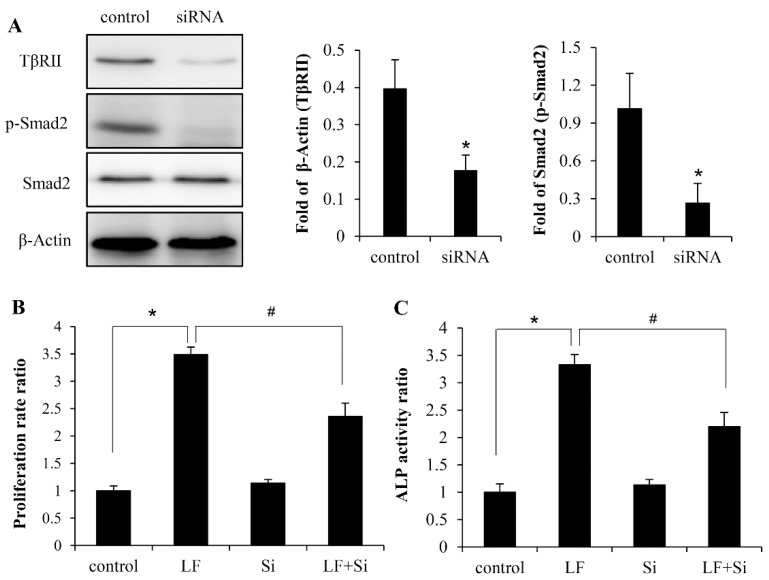
Inhibition of TβRII expression attenuated LF-induced osteogenic activity in C3H10T1/2 cells. The knockdown effect of TβRII siRNA on TβRII expression and the phosphorylation level of LF-induced p-Smad2 was determined (**A**). After pretreating cells with siRNA, LF (100 μg/mL)-induced proliferation (**B**) and differentiation (**C**) were detected. Values are means ± SDs, *n* = 3 (means of 5 replicates). * *p* < 0.05 compared with the control group; # *p* <0.05 compared with the LF + Si–treated group.

**Figure 5 ijms-20-02880-f005:**
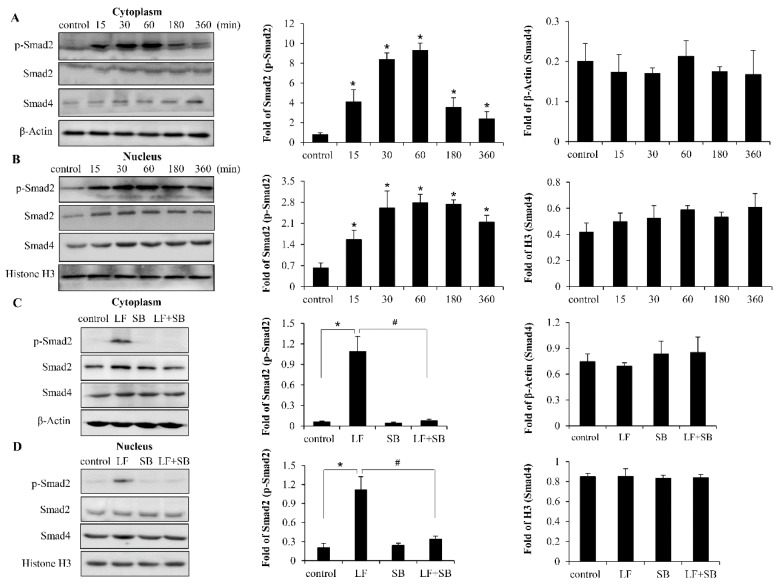
Western blot and the gray analysis showing activation of TGF-β/Smad2 signaling pathway by LF treatment in C3H10T1/2 cells. (**A**,**B**) Time-course analysis of Smad2 phosphorylation and nuclear translocation after treatment with 100 μg/mL LF. (**C**,**D**) Pre-treating cells with SB attenuates LF (100 μg/mL)-induced Smad2 and Smad4 activation. Values are mean ± SDs, *n* = 3. (* *p* < 0.05 versus control group; # *p* < 0.05 versus LF+SB treated group).

**Figure 6 ijms-20-02880-f006:**
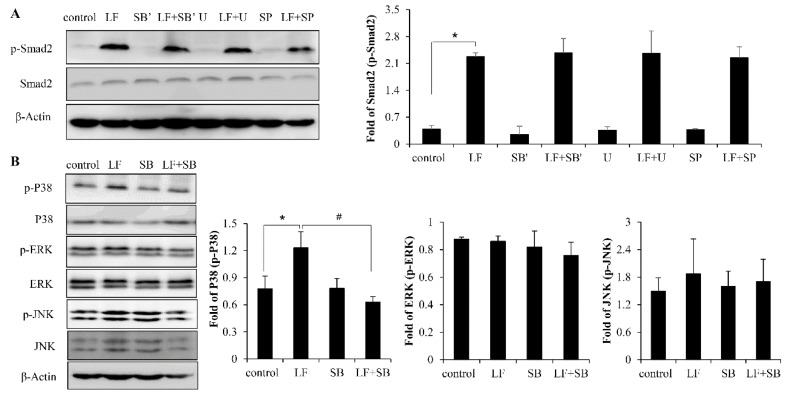
The crosstalk effect between LF-induced activation of TGF-β canonical and noncanonical signaling. (**A**) Pre-treating cells with inhibitors cannot attenuate LF (100 μg/mL)-induced Smad2 activation in MSCs. (**B**) After pretreating cells with SB, LF (100 μg/mL)-induced phosphorylation P38, ERK1/2, and JNK activation were detected. Values are mean ± SDs, *n* = 3. (* *p* < 0.05 versus control group; ^#^
*p* < 0.05 versus LF+SB treated group).

**Table 1 ijms-20-02880-t001:** siRNA knockdown of TβRII expression.

Gene Name	Primers Sequences
***OCN***	forward 5′-TGCTTGTGACGAGCTATCAG-3′ reverse 5′-GAGGACAGGGAGGATCAAGT-3′
***OPN***	forward 5′-ATCTCACCATTCGGATGAGTCT-3′ reverse 5′-TGTAGGGACGATTGGAGTGAAA-3′
***Col2a1***	forward 5′-GGGTCACAGAGGTTACCCAG-3′ reverse 5′- ACCAGGGGAACCACTCTCAC-3′
***FGF2***	forward 5′-GCGACCCACACGTCAAACTA-3′ reverse 5′-CCGTCCATCTTCCTTCATAGC-3′
***GAPDH***	forward 5′-TGGCAAAGTGGAGATTGTTGC-3′ reverse 5′-AAGATGGTGATGGGCTTCCCG-3′

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
