# Peer review of "Activation of TGF-β Canonical and Noncanonical Signaling in Bovine Lactoferrin-Induced Osteogenic Activity of C3H10T1/2 Mesenchymal Stem Cells"

_ijms, 2019, doi:10.3390/ijms20122880_

Reviewer 1 Report

Manuscript ID: ijms-506578

Title: Activation of TGF-β canonical and noncanonical Signaling in Lactoferrin-induced Osteogenic Activity of C3H10T1/2 Mesenchymal stem cells

Comments to Editor and authors

In the manuscript entitled “Activation of TGF-β canonical and noncanonical Signaling in Lactoferrin-induced Osteogenic Activity of C3H10T1/2 Mesenchymal stem cells” the authors describe very well the osteogenic activity induced by boovine lactoferrin and mouse mesenchymal stem cells C3H10T1/2.In the introduction the authors could update the bibliography  on lactoferrin functions (in addiction to Refs#3,4), given that there are extensive and recent reviews such as: 

Giansanti F, Panella G, Leboffe L, Antonini G. Lactoferrin from Milk: Nutraceutical and Pharmacological Properties. Pharmaceuticals (Basel). 2016 Sep 27;9(4). pii: E61. Review.

Amini AA, Nair LS. Lactoferrin: a biologically active molecule for bone regeneration. Curr Med Chem. 2011;18(8):1220-9.

In principle the model is valid but it would be much better to prove the effect of human lactoferrin on human mesenchymal cell lines optimizing the a homologous model. Moreover in the title it would be better to specify that the mesenchymal cells are mice and lactoferrin is bovine. 
Very interesting is the observation of the reduction of the expression of gene markers for the osteogenic activity in the presence of SB431542 before Lf treatment (pages 3-4, Figure 2C and 2D). It would be interesting to see if the inhibitory effect remains even by reversing the treatments, otherwise for first the Lf and then the inhibitor SB431542. 
In the experiments regarding the levels of p-Smad2 and nuclear translocation are decreased by the addition of SB431542 (Figure 5C, D) the indication of the times of tratments with the inibithor SB431542 either in the figure or in the text are not indicated, please indicate. 

For these reasons, after the corrections/additions requested the manuscript can be accepted.

Minor revisions:

Page 1, lanes 33 and 35: please write in vivo and in vitro in italics

Page 8, lane224: TGF-b please correct with TGF-b

Page 8, lanes 238, 239, 241, 243 and 262: please write in vivo and in vitro in italics

Author Response

Response to Reviewer 1:

1. In the manuscript entitled “Activation of TGF-β canonical and noncanonical Signaling in Lactoferrin-induced Osteogenic Activity of C3H10T1/2 Mesenchymal stem cells” the authors describe very well the osteogenic activity induced by bovine lactoferrin and mouse mesenchymal stem cells C3H10T1/2.In the introduction the authors could update the bibliography on lactoferrin functions (in addiction to Refs#3,4), given that there are extensive and recent reviews such as:

Giansanti F, Panella G, Leboffe L, Antonini G. Lactoferrin from Milk: Nutraceutical and Pharmacological Properties. Pharmaceuticals (Basel). 2016 Sep 27;9(4). pii: E61. Review.

Amini AA, Nair LS. Lactoferrin: a biologically active molecule for bone regeneration. Curr Med Chem. 2011;18(8):1220-9.

Response: Thank you for your comment. We updated the references recommended by reviewers instead of Reference 6. In addition, we replaced to the recent reference on lactoferrin functions in Introduction (Page 10 Reference 9, 12, 13, 17).

2. In principle the model is valid but it would be much better to prove the effect of human lactoferrin on human mesenchymal cell lines optimizing the a homologous model. Moreover in the title it would be better to specify that the mesenchymal cells are mice and lactoferrin is bovine.

Response: Thank you for your suggestion. It is a pity that human lactoferrin and human MSCs can not readily available. However, a high homology had been demonstrated between the human and bovine forms of lactoferrin. And the study suggested that supplementation of infant formulas with bovine LF may provide similar protection with the use of human milk [1].

Moreover, we added “Bovine” of the title. In addition, the name of “C3H10T1/2” represented that the cells derived from mouse.

3. Very interesting is the observation of the reduction of the expression of gene markers for the osteogenic activity in the presence of SB431542 before Lf treatment (pages 3-4, Figure 2C and 2D). It would be interesting to see if the inhibitory effect remains even by reversing the treatments, otherwise for first the Lf and then the inhibitor SB431542.

Response: Thank you for your comment. It is interesting to see the inhibitory effect remains even by reversing the treatments. However, the purpose of the study is to prove whether TGF-β pathway is involved in LF-induced upregulation of osteogenic factors. Thus, we had to inhibit the pathway with SB431542, then the inhibitory effect of the LF-induced upregulation was detected.

4. In the experiments regarding the levels of p-Smad2 and nuclear translocation are decreased by the addition of SB431542 (Figure 5C, D) the indication of the times of tratments with the inibithor SB431542 either in the figure or in the text are not indicated, please indicate.

Response: Thank you for pointing this out. We added “by up to 100%” in the text (Page 6, line 163) to indicate the times of treatments with the inhibitor SB431542.

5Page 1, lanes 33 and 35: please write in vivo and in vitro in italics

Response: Thank you for your comment. We wrote “in vivo” and “in vitro” in italics on line 33 and 35. Similarly, we have also modified "in vivo" and “in vitro” in the whole paper.

6Page 8, lane224: TGF-b please correct with TGF-b

Response: Thank you very much for your careful review. We corrected “TGF-b” to “TGF-β” on line 224 in the revised manuscript.

7Page 8, lanes 238, 239, 241, 243 and 262: please write in vivo and in vitro in italics

Response: Thank you for your comment. We wrote “in vivo” and “in vitro” in italics on line 239, 240, 242, 244, and 263 in revised manuscript.

Reference

1.             Bhatia, J., Bovine Lactoferrin, Human Lactoferrin, and Bioactivity. J Pediatr Gastr Nutr 2011, 53, (6), 589-589.

Reviewer 2 Report

In this study, the authors examined the involvement of TGF-β/smad2 signaling in Lactoferin-induced osteogenic activity using C3H10T1/2 mesenchymal stem cells, following their previous study (J. Nutr 2018, 148(8) 1285-92). Although this study is well performed, there remain some concerns that have not been adequately addressed.

Comments

1. In this study, the authors revealed the effect of lactferin on the differentiation of C3H10T1/2 cells in Figure 1 using ALP activity. Is this study first evidence on osteogenesis in MSCs? In this case, only ALP activity was not enough. The morphological changes or Alizarin red staining in MSCs should be demonstrated, if the authors suggest the osteogenic differentiation of MSCs. And also, the authors only showed the ratio in MTT assay and ALP activity in this study. Please show the real data on each experiment.

2.  The authors revealed the mRNA expression of TGFβRI and II in LF-induced MSCs in figure 3. Hoe was on the protein level? It needs to show the protein expression of TGFβR I and II.

3. The author showed Western blotting data to the protein expression of p-smad2 and MAPK in this study, and also folds change of protein level using β-actin. In these experiments, the author should show the expression of total smad2 and total p38, ERK, JNK in figure 4, 5, and 6. In addition, please use total protein, not β-actin, when the authors try to show the fold change of protein expression in these experiments.

4. Figure 4 and 6 suggested that TGF-β and MAPK signaling had the effect of LF-induced signaling. In figure 4, ALP activity was showed in TGF-βRII knockdown MSC cells. How was the ALP activity or gene expression of osteogenic factors in MAPK inhibitor-treated MSCs?

5. Why use siRNA for TGF-βRII and SB for TGFβRI in this study? Previous the author’s paper has done same experiment design. However, it needs to clearly describe in discussion sections.

6. The several abbreviations for symbol are not defined in abstract.

7. Please showed time course in Figure2.

Author Response

Please refer to the Word. Thanks.

Round  2

Reviewer 1 Report

After the revision the authors followed the indications and the answers are exhaustive. For these reasons the work can be accepted in the present form

Reviewer 2 Report

This revised version is well written manuscripts.